# A qualitative study to examine hidden care burden for older adults with overweight and obesity in England

Gargi Ghosh[1]*, Hafiz T. A. Khan[2]*, Salim Vohra[2]*

1 Faculty of Health, Medicine & Social Care, School of Nursing and Midwifery, Anglia Ruskin University, Chelmsford, Essex, United Kingdom, 2 Public Health Group, College of Nursing, Midwifery and Healthcare, University of West London, Brentford, United Kingdom

* gargi.ghosh@aru.ac.uk (GG); hafiz.khan@uwl.ac.uk (HTAK); salim.vohra@uwl.ac.uk (SV)

## Abstract

### Objectives

The study aimed to explore the phenomena related to formal and informal social care needs among overweight and obese older adults living in England.

### Background

Despite the rising prevalence of obesity among older adults, its impact on social care needs remains underexplored. Existing research highlights significant unmet social care needs among older adults, yet the specific challenges faced by those who are overweight or obese have received limited attention. This study addresses this gap by exploring and understanding the social care experiences and needs of older adults in England who are overweight or obese.

### Methods

Participants were recruited from a local National Health Service (NHS) health centre in London England using a purposive sampling strategy to the point of analytical saturation. A total of 45 participants were invited and of these 33 participants were eligible to take part. All participants in this study are either of British origin or immigrants to the UK from various nationalities. A semi-structured interview was conducted, and a qualitative structural narrative analysis was undertaken.

### Results

The study found that older adults, who are overweight or obese, were more likely to have physical health problems and problems with mobility. They were more likely to have informal voluntary care and support rather than formal social care support. They also had a weaker social support network, were more isolated and frustrated, lacked housing adaptations, felt unsafe, felt they were a burden to their families and felt discriminated against by the wider community. Care and support needs if not met, then these are likely to generate or widen health inequalities over time.

**Data availability statement:** The original transcription cannot be submitted without participant consent; however, a version with all identifiable details removed has been provided within the manuscript in Table 2. All relevant data are within the manuscript and its Supporting Information files.

**Funding:** The author(s) received no specific funding for this work.

**Competing interests:** The authors have declared that no competing interests exist.

## Conclusions

This study provides a unique perspective on unmet care needs among overweight and obese older adults in England. It highlights the compounded challenges faced by this population, emphasising the importance of holistic social care approaches that address both health and psychosocial needs. Findings suggest that minimal yet targeted interventions, such as accessible support networks and public health policies promoting social engagement, could significantly improve wellbeing and reduce long-term health inequalities.

## Introduction

As the global population ages, the prevalence of chronic conditions, including multi-morbidity, is increasing, posing significant challenges to healthcare systems [1,2]. In Great Britain, 36% of adults report having a disability, and 20% experience other long-term limiting conditions [3]. Obesity is a significant contributor to morbidity and mortality, responsible for more than 3.4 million deaths worldwide, 4% of life years lost, and at least 4% of disability-adjusted life years (DALYs) [4]. Socioeconomic disparities further exacerbate the issue, with obesity prevalence nearly double in the most deprived areas compared to the least, at 36% and 20%, respectively [5].

Despite growing concerns over obesity's impact on health, its influence on social care needs among older adults remains underexplored [6,7]. Older adults with obesity often experience functional impairments that increase their dependence on personal care assistance [8]. Research has shown that morbidly obese older women require significantly more support with daily living activities than their normal-weight counterparts [9]. Beyond physical challenges, obesity in older adults is frequently associated with social discrimination, stigma, and body image concerns, all of which can negatively impact mental wellbeing, self-esteem, and overall health behaviours [10–13]. Such experiences may further compound the complexity of social care needs for this population. Wharton et al. [11] identified weight bias in healthcare settings as a key factor contributing to delayed medical care, which can exacerbate health conditions and increase dependence on social support services. Furthermore, internalised weight stigma among older adults has been linked to poorer psychological wellbeing and reduced engagement in health-promoting behaviours [12].

Social care encompasses a broad range of support services that help individuals maintain independence and quality of life. These needs are typically met through formal state-funded services, private care arrangements, or informal family support [14,15]. However, unmet social care needs remain a persistent issue among older adults, particularly those with chronic health conditions. The Health Survey for England (HSE) reported that a significant proportion of older adults require more care than they currently receive, with unmet needs affecting daily activities such as washing, dressing, and mobility [16,17]. Studies indicate that approximately 47% of care recipients believe they require additional services to adequately meet their needs [18,19]. Additionally, NHS England (NHSE) data suggests that 19% of men and 28% of women over the age of 65 struggle with at least one activity of daily living (ADL), with 12% of men and 15% of women reporting unmet needs for at least one instrumental ADL (IADL) [20,21].

In England, access to formal state social care is contingent on specific eligibility criteria, including financial means, physical and mental health status, and the availability of informal caregivers [22,23]. Budgetary constraints and policy changes have further restricted access to state-funded support, leading to an increasing reliance on informal family carers and private services [16,24]. This reliance places a growing burden on unpaid carers, who may themselves

experience physical, emotional, and financial strain. As a result, gaps in care provision can contribute to increased hospital admissions and poor health outcomes for older adults with unmet needs [25].

The Care Act [26] highlights the importance of identifying and addressing unmet social care needs through preventive strategies that delay or reduce the need for intensive care support. These include primary prevention measures aimed at individuals without existing social care needs, as well as secondary and tertiary prevention efforts focused on those already receiving care. Moreover, the Care Act Guidance on Secondary Prevention [27] emphasises the need to address "low-level" care needs early to prevent the escalation of long-term support requirements.

Given the rising prevalence of obesity among older adults and the growing challenges within the social care system, this study aims to explore and understand the social care experiences and needs of older adults in England who are overweight or obese. By investigating their unmet social care needs, the study seeks to identify the hidden burden of disease, the associated healthcare costs, and potential strategies to enhance quality of life. A report by Local Government Association (LGA) [2] informed that the Equality Act of 2010, is yet to be safeguarded by a person's weight category. However, helping a person to overcome societal impairments have an association with significant cost implications and workforce. Hence it demands an improvised public health policy and planning. Addressing these gaps is critical for optimising the effectiveness of formal and informal social care support and ensuring that older adults receive the necessary assistance to maintain their independence and wellbeing.

## Methods

This study applies a qualitative methodology to explore older adults' perspectives on unmet social care needs, generating insights into the types and levels of support required rather than estimating the prevalence of unmet needs within the population. Therefore, the study has undertaken a purposive sampling strategy, using semi-structured, one-to-one interviews to explore the experiences of participants who met the study's inclusion and exclusion criteria. Each participant received an information sheet and consent form prior to the interview. Sampling was done to the point of redundancy. A single NHS General Practitioner (GP) surgery was selected through convenience sampling, providing a private and accessible setting conducive to in-depth discussions. A loosely structured interview guide, based on a validated English Longitudinal Study of Ageing (ELSA) questionnaire (freely available), was used to ensure effective use of time. Open-ended and probing questions adapted to participants' responses explored weight-related barriers in disability, health, wellbeing, and social care support. The interview protocol was pilot tested with the first two participants to assess clarity, accuracy, and effectiveness. This process evaluated recruitment, consent, interview questions, and recording procedures. The pilot also helped determine pacing and interview duration, leading to refinements in the final questions. All participants were debriefed after the completion of the interviews.

### Ethical statement

Ethical approval was granted by the University of West London's CNMH Research Ethics Panel and the UK Health Research Authority (HRA) (Protocol No: 21374279) in 2018. In compliance with UK regulations, ethical approval was obtained through the HRA's Integrated Research Application System (IRAS Project ID: 253586). The Research Ethics Committee (REC Reference: 19/LO/1093) reviewed the application, and HRA and Health and Care Research Wales (HCRW) granted approval.

All participants provided written informed consent, agreeing to the use of their anonymised interview transcripts for teaching, research, and publication. Identifiable details were removed, and pseudonyms were assigned. Data collection took place between July 3, 2020, and December 1, 2020.

## Underpinning theory

This study draws upon the life course approach and social cognitive theory to contextualise obesity among older adults. The life course approach explains how cumulative social, economic, and environmental factors influence health across different life stages, necessitating proactive public health interventions [28,29]. The social cognitive theory emphasises the interaction between personal experiences, environmental influences, and learned behaviours in shaping health outcomes [30,31].

Moreover, the study incorporates the social model of disability and a human rights perspective to highlight structural barriers that contribute to dependency and impairment. Wellbeing is framed as recognising individuals' abilities rather than disabilities, emphasising empowerment and independence [32,33]. Finally, the concept of 'need' is examined using Bradshaw's taxonomy, distinguishing normative, felt, expressed, and comparative needs [14].

## Defining the key phenomena

**Obesity.** To determine unmet care needs, weight was categorised into five groups: normal weight, overweight, moderate or class I obesity, severe or class II obesity, and morbid or class III obesity. This was based on the three categories of obesity developed by World Health Organisation (WHO): obesity class I (BMI = 30-34.9 kg/m2), obesity class II (BMI = 35-39.9 kg/m2) and obesity class III (BMI ≥ 40 kg/m2) [34].

**Unmet care needs for social care and support.** This study defines unmet care needs as any identified needs, whether related to functional or mobility impairments or broader social and emotional aspects, that are not adequately supported by existing social care services. Unlike narrower definitions that focus primarily on ADLs and IADLs, this study expands the concept to encompass social contact, safety, loneliness, and a sense of purpose, thereby capturing hidden unmet needs.

Vlachantoni et al. [16] describe unmet care needs as the gap between demand (such as mobility impairments or difficulties in performing ADL/IADL tasks) and the supply of care, whether informal (family), formal (public/state services), or private (paid care). Dunatchik et al. [20] caution that a purely task-based assessment may overlook critical aspects of wellbeing, such as social engagement and emotional fulfilment. The Care Act [26] reinforces this preventive approach, advocating for proactive measures to mitigate unmet needs before they result in greater dependency on social care services.

**Conceptual model of unmet care needs.** To achieve the study's objectives, a conceptual model (see Fig 1) was developed to identify gaps in social care provision, particularly among individuals who either do not meet formal eligibility criteria or are unaware of their own unmet needs. By predicting a range of required support services, the model aims to inform policy decisions and resource allocation for more inclusive care planning. While this approach may lead to an overestimation of unmet needs, identifying potential care gaps at an early stage allows for strategic intervention, ultimately reducing long-term health and social care costs.

In this model (Fig 1), the Y-axis denotes social care support (S), and the X-axis refers to social care needs (N). According to the model, from the centre (0,0), social care support increases along the Y-axis, and social care needs increase along the X-axis. Therefore, according to the need and support received, the population are divided into five groups. Group A

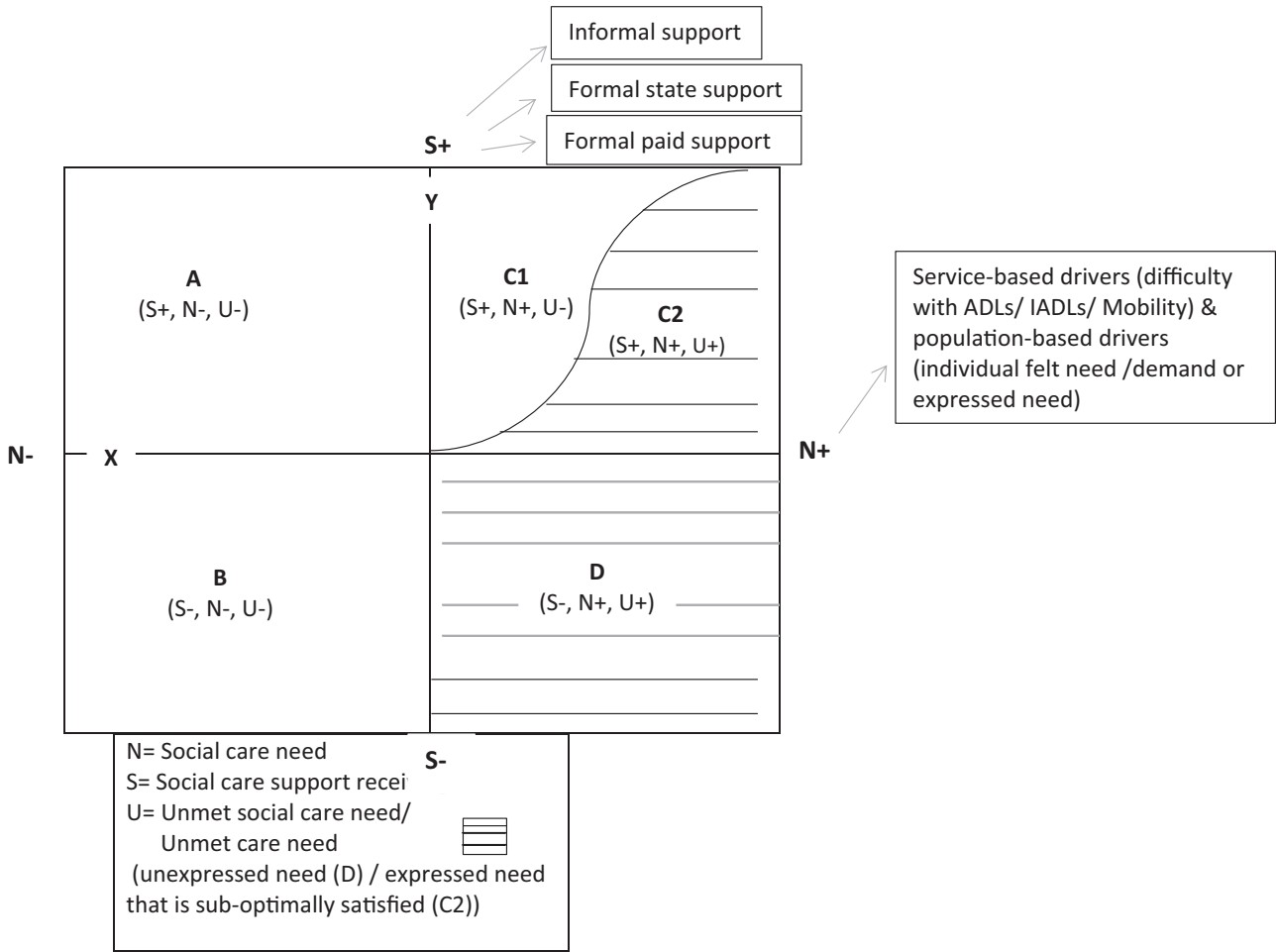

**Fig 1. Conceptualised model on unmet care needs for social care and support.** Constructed by Ghosh [45].

represents older adults who do not have any need (in terms of functional and mobility impairments and any other individual felt needs), but they receive support (whether informal family, formal state or formal private paid-for care or a combination). Group B represents the older adults who do not have any needs and are not receiving any social care. Group C1 represents older adults who have at least one need (either social care service-defined or individual-felt), and their needs are fulfilled by the social care they receive. Group C2 represents older adults who are not satisfied with the support they are receiving according to their felt needs. Hence, they have existing unmet felt needs and unmet needs for social care. Group D represents older adults who have at least one need but are not receiving any social care. Therefore, this group of people has unmet felt needs and unmet needs for social care. The present study's focus is to collect primary data about two groups of older adults in the mode, groups C2 and D, and explore the association between unmet felt needs and unmet needs for social care by the degree of BMI in older adults.

## Eligibility criteria

Participants were recruited based on the following criteria:

Inclusion: Adults aged 50 and over, BMI ≥ 18.5 kg/m², English-speaking, visiting minor illness clinics, and capable of providing informed consent.

Exclusion: Non-English speakers, individuals with advanced dementia, severe/profound intellectual disabilities, or genetic syndromes affecting weight (e.g., Prader-Willi, Cohen, Bardet-Biedl)

### Data collection and analysis

Semi-structured interviews lasting 30-40 minutes were conducted in a private GP surgery setting. Out of the total 33 cohorts, 5 participants requested a phone interview. All phone interviews were carried out using the GP surgery's phone. Interviews were audio-recorded and transcribed verbatim. Field notes and reflective diaries supplemented the data collection. A structural narrative analysis procedure was adopted and applied, where each narrative was structured with the five common chapters in the chronological move towards the unmet care need. Each narrative started with the character introduction which was followed by life in terms of disability, health status and life satisfaction, existing care and support, concluding the interview with whether the interviewee wants to add anything more for their own care and support and finally, the results of unmet social care/support needs. The term 'care and support' was consistently used to assess gaps between demand and received services. After each interview, the recording device was turned off, and participants were invited to ask questions. They were also reminded of how the study findings would be disseminated.

## Results

Research findings are discussed under three headings: participant background profile, a summary of the narrative discussion, and interpretation of the narrative. Specific cases are purposefully selected for narratives according to the study objective. The interviews that answered the research question and provided the maximum information related to the study objective were chosen. The discussion of 'unmet care need' is framed around the framework (Fig 1); a detailed description of the framework is presented above.

### Participant's profile

Table 1 outlines each participant's demographic and background information and anthropometric height and weight measurements, calculated BMI, and assigned weight category. Participants' weight category is assigned according to the WHO BMI classification, as discussed above.

### The main outcome of narrative analysis

A summary of 10 older adults (aged 50+ years), where each of them experienced an 'unmet care need', is presented below in Table 2. All names in the narratives are pseudonyms. The unmet care needs of each participant are in bold. In the narratives, symbols like: (.) represent pauses and […] represents omitted materials in the conversation, and italic words indicate the direct quotes from participants' talk, the exact way they told their narratives.

According to the conceptual framework (Fig 1), all the participants in the narratives in Table 2 either had 'unexpressed needs' (D) or 'expressed needs that were sub-optimally satisfied' (C2). The 'unexpressed needs', where participants either had care needs in terms of ADLs, IADLs, and mobility but did not perceive their difficulties (case numbers- 19, 14) or participants were reluctant to request help and support (case numbers- 19, 27, 21, 14, 32). The 'expressed needs that were sub-optimally satisfied', where participants either not qualified to meet the eligibility criteria to access the local authority social care support (case number- 10, 7, 14) or had inadequate existing social care (case number- 2, 27, 30, 32, 19) and support. Some participants also had both unexpressed needs and expressed sub-optimally satisfying needs (case numbers- 27, 14, 32, 19).

**Table 1. Anonymised demographic details of participants.**

| Interviewees by case number | Age | Gender | Ethnicity | Marital status/existing partner | Height (m) | Weight (kg) | BMI (kg/m2) | Weight status (WHO categorised) |
|---|---|---|---|---|---|---|---|---|
| 1 | 78 | Male | British | Yes | 1.71 | 76 | 26 | Overweight |
| 2 | 50 | Female | Indian/ Mauritian | Yes | 1.65 | 115 | 42.4 | Class III obesity |
| 3 | 66 | Male | Bulgarian | Yes | 1.75 | 57 | 18.5 | Normal |
| 4 | 50 | Male | Russian | Yes | 1.70 | 60 | 23.2 | Normal |
| 5 | 60 | Male | British | Yes | 1.78 | 78 | 24.6 | Normal |
| 6 | 60 | Female | British | No | 1.54 | 56 | 23.6 | Normal |
| 7 | 57 | Male | Pakistani | No | 1.80 | 105 | 32.4 | Class I obesity |
| 8 | 58 | Male | Indian | Yes | 1.73 | 74 | 24.7 | Normal |
| 9 | 53 | Female | African | Yes | 1.60 | 102 | 39.8 | Class II obesity |
| 10 | 52 | Female | British | No | 1.60 | 102 | 39.8 | Class II obesity |
| 11 | 61 | Female | British | Yes | 1.55 | 105 | 43.7 | Class III obesity |
| 12 | 57 | Female | Indian | Yes | 1.52 | 79 | 26 | Overweight |
| 13 | 63 | Male | British | Yes | 1.75 | 87 | 28.4 | Overweight |
| 14 | 51 | Male | Spanish | Yes | 1.82 | 131 | 39.6 | Class II obesity |
| 15 | 75 | Male | Greek | Yes | 1.85 | 100 | 29.2 | Overweight |
| 16 | 58 | Female | British | Yes | 1.77 | 105 | 33.5 | Class I obesity |
| 17 | 59 | Male | Pakistani | Yes | 1.61 | 67 | 25.8 | Overweight |
| 18 | 62 | Female | British | Yes | 1.58 | 91 | 36.5 | Class II obesity |
| 19 | 66 | Male | British | Yes | 1.91 | 102 | 28 | Overweight |
| 20 | 52 | Female | British | Yes | 1.65 | 112 | 41.1 | Class III obesity |
| 21 | 73 | Male | British | No | 1.71 | 96 | 32.8 | Class I obesity |
| 22 | 55 | Female | British | Yes | 1.70 | 78 | 27 | Overweight |
| 23 | 54 | Female | British | Yes | 1.62 | 77 | 29.3 | Overweight |
| 24 | 69 | Male | British | Yes | 1.68 | 87 | 30.8 | Class I obesity |
| 25 | 60 | Male | British | No | 1.67 | 83 | 29.8 | Overweight |
| 26 | 69 | Male | British | No | 1.73 | 80 | 26.7 | Overweight |
| 27 | 52 | Female | British | Yes | 1.52 | 67 | 29 | Overweight |
| 28 | 63 | Female | Italian | No | 1.46 | 71 | 33.3 | Class I obesity |
| 29 | 56 | Female | British | No | 1.65 | 78 | 28.6 | Overweight |
| 30 | 57 | Male | British | Yes | 1.76 | 95 | 30.7 | Class I obesity |
| 31 | 74 | Male | British | Yes | 1.76 | 97 | 31.3 | Class I obesity |
| 32 | 53 | Female | British | Yes | 1.61 | 93 | 35.9 | Class II obesity |
| 33 | 55 | Male | British | Yes | 1.79 | 85 | 26.5 | Overweight |

The participants with impaired mobility or problems in daily living faced the following challenges of unmet needs: ill health, loneliness, lack of socialisation, lack of emotional support (felt frustrated, depressed, burdened, guilty, embarrassed and anxious), felt vulnerable due to lack of carers' time (unpaid carers), lack of carers' knowledge, lack of housing adaptations, lack of support with everyday activities, lack of self-confidence obtaining existing social care services, lack of safety in one's own house and lack of financial aid (individual raised).

From the above narratives, it is discerned that most of the participants mentioned their painful back and joints resulting from different types of arthritis (case number- 2, 27, 53, 57), as the main reason for their disability, or the pain due to cancer, kidney dysfunction and previous surgery.

Participants with higher obesity levels reported unmet needs primarily linked to emotional distress, including stress, frustration, social discrimination, isolation, feeling like a burden,

**Table 2. Narrative analysis findings.**

| Case No. | Name and demographics | Narrative analysis |
|---|---|---|
| 2 | **Sunita (50 years, female, morbidly obese, positive cohabitation)** Sunita is a 50-year-old female, who lives with her husband. She is from an Indian/ Mauritian ethnic background and has morbid obesity (BMI = 42.4 kg/m2). | <u>Life in terms of disability, health status and life satisfaction</u>: Sunita faces difficulty with several tasks in daily living, such as using the toilet, including getting up or down, preparing a hot meal, working around the house and garden, and climbing stairs due to her painful back and joints. However, she does not use any kind of mobility aids, and according to her, *I don't think I am that old*. She considers her health status as 'fair', however, does not consider that she has any type of chronic illness or mental impairment (health status): *Generally, if you are good, you don't need medication, you don't […] you don't be in pain every time you wake up in the morning […] it just basically depends on the days, some days are good, but most days are like on a scale of 1 to 10 (higher indicates better health status), I would just call it like 5 and sometimes lower than this [...] it just depends on how you spend the night and […] I don't sleep […], so it just depends really.* She feels safe in her own home and on a scale of 0 to 10 (higher indicates safer), she marks it 9. But she does not like being staying on her own (safety in the home): *Well, if I remove the panic attack […], you know when you are ill (.) sometimes you feel you don't want to be alone […] because I was always surrounded by all my family […] especially if you are not feeling well. You start thinking of the worst (.) you know, so it just depends (.) so what I think about at home, if I am well surrounded, then it is a good 9, but if I am alone, I always panic (.) so I would say 4 (.) so depends on the time when he (husband) comes back home (.) I feel safe (.) I feel happy.* Sunita considers herself 'unsatisfied' in terms of her overall life satisfaction, this is mainly due to the COVID-19 pandemic that has worsened the work situation and lets her be socially isolated. However, if not for the pandemic, she would have considered her life satisfaction as 'partially satisfied' (life satisfaction): *Because of my health (.) fully satisfied would be someone, who does not need any support (.) you know they can rely on themselves, so they can do whatever they want in a way, but with my health and what is going on around (.) plus the pandemic makes worse.* <u>Existing care and support</u>: She has installed a speciality toilet (housing adaptation). She gets a "lot of support" from her husband, and she considers herself "very lucky". However, she feels her needs are 'usually met', though the amount of time her husband spends with her is not enough (care and support/ unmet need): *He is the type of person who understands one word; you know if you say 5 words together, then the problem is (.) he doesn't have ears (giggles) […]. Still, he has to work […] because you know there is no flexibility from work; you know if you have support from a work (.) that's what the problem is at the moment […]. Still, they are not giving any support from his end to be able to provide the care that I need (.), so in a way, that's where the problem is (.), and you can't lose your job, especially of this time […], so he doesn't have the support needed to be able to support me […] sometimes I do need somebody to be around during the day (.) then either I have to go to a hospital or call a friend.* Sunita is 'very satisfied' with how she presents herself in terms of her appearance and cleanliness. She is not so happy about the food and drink she gets at home (care and support): *"he (husband) doesn't know how to cook […] because he always thinks that I need to lose weight and gives me healthy food (.) I can't really follow healthy diet".* However, the way her husband treats her makes her feel excellent about herself. <u>Concluding the interview with whether the interviewee wants to add anything more for his/her own care and support</u>: Sunita wishes (unmet need): *"may be having someone to speak to (.) having someone to listen (.) and may be having a priority to access to a facility (.) you know when you are on your own and you don't have help (.) you need to call a place like hospital or book something […] for your own medication or something, you have to stay on the phone and at the same time you have to look after yourself (.) so for these things […] having a priority […] like giving a number […] and the third thing I believe that's more important is may be somebody like (.) more like a health care support or assistant just for cooking (.) not cooking, but food or […] may be just someone can come and see me (.) may be for an hour just to say hello (.) if I am not well, just to say (.) can I call an ambulance? (.) The person doesn't have to come with me but at least help me to get access to all the facilities […] may be a carer with compassion".* <u>The result on unmet social care/support needs</u>: From the above narrative, it is identified that Sunita feels vulnerable because her primary carer (her husband) is unable to spend the amount of time (**carer's time**) she feels is needed. **Loneliness** is one of the significant unmet needs that Sunita has experienced and sometimes that triggers panic attacks (**lack of emotional support). The lack of safety in her own home** is another concern likely associated with being alone in the house when her husband is at work and her family is not around. Moreover, Sunita **lacks the self-confidence to access existing social care services,** as she does not consider herself old enough to access certain types of support (mobility aids) and desires to have a more accessible support service. Furthermore, the foods she gets in everyday life do not match her wishes (**unmet want**). |
| 27 | **Lucy (52 years, female, overweight, positive cohabitation)** Lucy is a 52-year-old female, who lives with her husband and is in full-time employment. Her children are all grown up and have moved away. Lucy is from a white British ethnic background and is overweight (BMI = 29 kg/m2). | <u>Life in terms of disability, health status and life satisfaction</u>: Lucy experiences several challenges for daily living, such as difficulty in walking, cutting up foods, getting in and out of bed, shopping for groceries, gets confused sometimes taking medications, working around the house and garden. However, she does not use any walking stick or any type of aid for moving. Lucy states her health status as 'fair', as she suffers from several conditions: osteoarthritis, erythromelalgia, and vitamin deficiency (the list is long) and, according to her, walking gives her pain (health status): *pain is what I struggle with the most because that (exercise machine) all seem to cause pain […] most of the time I am fairly good (.) I am quite a sort of up bit and happy.* Lucy feels very safe in her own home, even when her husband is not at home (safety in the home). Lucy is 'fully satisfied' with her life overall (life satisfaction) *[…] there are people far worse than me.* |

*(Continued)*

**Table 2.** (Continued)

| Case No. | Name and demographics | Narrative analysis |
|---|---|---|
| | | Existing care and support: Lucy's husband takes care of her besides being in full-time employment. Moreover, Lucy's husband is very supportive, as she says, *looks after me well*. Currently, he (husband) does not receive any care allowances. But taking care and support from her husband makes Lucy 'partially satisfied' (care and support):<br>*Sometimes I feel guilty (.) because I think he* (Lucy's husband) *was working hard as well, then he comes back home, he does the dinner […] washing, ironing, gardening* (chuckles) *he does everything. Although I have recently said to him (.) can you not do it? If you leave something for me (.) so that I don't become a vegetable (stressed the word) (.) sitting on the sofa doing nothing!* (Long pause) *He says he does it because he doesn't like to see me in pain.*<br>Moreover, she considers her needs are 'usually met' and the amount of time her husband spends with her is not enough, as she informs, *I wish it could be more* (care and support):<br>*I think I get by quite well (.) I think the only time I struggle mainly […] if I am going out and walking […] if I want to go to the shop […], I get put off […] I think you know (.) I got to walk from the car to the shop […] or going to the car park […] I have toyed with (.) may be applying for a badge thing* (disability badge) *then I get put off […] because then I think I can still walk, but (.) there are a lot of people out there that are more deserving of it […] I just think to myself […] at what point (.) do I give into that or (.) people are going to look at me (.) and think she doesn't need that.*<br>However, Lucy is 'fully satisfied' with her appearance, cleanliness, maintaining personal hygiene and the food and drinks she gets in everyday life.<br>But she considers that the help and support she receives from her husband makes her feel 'poor' (with a chuckle) about herself. She said that it makes her feel "*bit guilty, lazy, a burden (.) I know that doesn't make any sense […] yeah, I feel burden! (.) and then I think 'right (.) stop thinking about that, move onto the next.*<br>Concluding the interview with whether the interviewee wants to add anything more for his/her own care and support:<br>Lucy wishes (unmet need):<br>*I only need help with (.) maybe with that […] badge thing* (disable badge)*, but that's about it […] I am not going to start going down (.) using frames and walking sticks (.) because then, that just draws attention to it (.) and I think 'no'* (chuckles).<br>The result on unmet social care/support needs: From the above narrative, it is noted that Lucy perceives herself as a burden and feels guilty (**lack of emotional support**) for the support and care she receives from her husband; besides, she desires for more of his (husband) time (**carer's time**). Moreover, Lucy is reluctant to request help and **hesitant to accept having unmet needs,** as she compares herself with others around her and feels appreciative of what she has. "These attitudes led some participants to deny or minimise their needs for care and support, and/or to normalise challenging situations" [41, p. 23]. She puts a high value on her independence as an important part of her self-esteem. In addition, Lucy is afraid of being **socially discriminated** against. Therefore, she gives more importance to other's perceptions of herself than her needs. |
| 30 | **David (57 years, male, moderately obese, positive cohabitation)**<br>David is a 57-year-old male, who lives with his wife. He is from a white British ethnic background and has moderate obesity (BMI = 30.7 kg/m2). | Life in terms of disability, health status and life satisfaction: David faces difficulty with almost all the activities of daily living. However, except with few activities that he considers himself capable of, such as communication, making phone calls and recognising physical danger. In addition, David uses several supporting aids that help him be independent, such as a walking stick, walker, wheelchair, and personal alarm for a fall. He suffers from several health conditions, including urinary incontinence, painful back due to lumbar decompression surgery. David considers his health status as 'poor' and is 'unsatisfied' in his overall life satisfaction (life satisfaction):<br>*I have depression […] I wasn't like this before (.) it's just too much suffering […] my wife changes my nappy (.) I can't help (.) makes me frustrated and depressed.*<br>David considers that he is not very safe in his own home and scored 5 (on a Likert scale of 0 to 10, where 10 is the safest). He also feels that *medication makes me drowsy (.) that I always feel weakness.*<br>Existing care and support: David had several housing adaptations. Currently, he is getting disability allowance and housing allowance. He sees his wife as his primary carer. David is concerned that his wife does not get any government allowances because she is from abroad (care and support):<br>*We applied for LA* (Local Authority) *allowances (.) we meet all the criteria, still (.) it's a long fight […] we need to sort out her (wife) immigration first […] she deserves it (.) lawyer is fighting the case.*<br>However, David is 'fully satisfied' with the care and support he receives from his wife in terms of his appearance, cleanliness and the food and drink he gets in everyday life. He says, *she is wonderful (.) she is always there.* The way his wife treats him makes him feel excellent about himself. But he considers his needs are 'sometimes met' (care and support):<br>*I wish to walk normally again (.) I start panicking with people […] don't like to mix with people* (agitated).<br>Concluding the interview with whether the interviewee wants to add anything more for his/her own care and support:<br>David expresses his worry about his wife's immigration. Since that would enable them to apply for the LA allowances. He concludes the interview with (unmet need):<br>*it would be helpful if I could have some economic support.*<br>The result on unmet social care/support needs: From the above narrative, it is noted that David is frustrated with his current situation and feels guilty about taking help from his wife (**lack of emotional support**). David expresses his **social anxiety** (mixing with others), along with **financial hardship**. However, he yearns to walk again, and it is apparent that he misses socialising. David judges having a **lack of safety in his own home**. |

*(Continued)*

**Table 2.** (Continued)

| Case No. | Name and demographics | Narrative analysis |
|---|---|---|
| 3 | **Ivan (66 years, male, normal weight, positive cohabitation)** Ivan is a married 67-year-old male who lives with his wife, son, and daughter in law. Ivan is from a white Bulgarian ethnic background and is normal weight (BMI = 18.5 kg/m2). | <u>Life in terms of disability, health status and life satisfaction</u>: Despite having a normal weight, Ivan encounters several challenges in daily living, such as difficulty in dressing, walking, getting in and out of bed, using the toilet and taking medication. He considers his health as 'fair'. Ivan is a bowel cancer patient, who had to go through a couple of surgeries but still suffers from severe pain and discomfort. Although he does not like to use any mobility aids and does not consider having any chronic illness or mental impairment (health status): *Because of all this pain (.) I am getting severely; I am actually not getting any care […]. I was calling into the GP […] but was not able to get through (.) to get me some actual physical help.* <u>Existing care and support</u>: Ivan's wife is the main person who looks after him. And Ivan is happy for the time she stays with him. Moreover, Ivan is satisfied with the food and drink that he gets every day and happy with his appearance and cleanliness. However, taking support from his wife makes him upset, he is 'unsatisfied' with this and considers himself a burden. The way he is cared for or treated by his wife makes him feel 'poor' about himself (life experience/care and support): *Obviously, my wife is caring (.) looks after me well […] but there are a lot of things that she is not enough (.) such as the medication, such as (.) knowing actually what the issue is with me (.) to manage it properly […] I need someone who has the knowledge to manage (.) someone who will stop me feeling embarrassed with my life (.) I was a very independent person […] in my culture, the husband should take care of his wife (.) not the other way round […] my wife is not well, she has osteoporosis.* His needs are not met fully, mainly due to the pain, which marks it as 'sometimes met'. According to him, *I need help (.) with my doctor to understand my problem* (care and support). <u>Concluding the interview with whether the interviewee wants to add anything more for his/her own care and support</u>: Ivan shows concern about his wife's health and wellbeing and mentions that she is helpless. As Ivan expresses his expectation (unmet need): *I guess I need to have some sort of help from the government for extra money or fund (.) maybe it could help me (.) because it costs a lot for a good healthy diet.* <u>The result on unmet social care/support needs</u>: From the above narrative, it is noted that Ivan is embarrassed by taking the support and care from his wife, who is already unwell and considers himself a burden **(lack of emotional support)**. In addition, Ivan feels vulnerable and shows concern that his wife lacks the necessary skills to provide him proper support **(lack of carer's knowledge)** besides his **financial hardship**. Moreover, his expressed needs are sub-optimally satisfied, and he is insecure about his existing unpaid care. Finally, he is having **lack self-confidence in obtaining existing social care services.** |
| 10 | **Lisa (52 years, female, severely obese, positive coresidents)** Lisa is a 52-year-old lady who is recently divorced and lives with her daughter. Lisa is from a white British ethnic background and has severe obesity (BMI = 39.8 kg/m2). | <u>Life in terms of disability, health status and life satisfaction</u>: Lisa is entirely self-sufficient and capable of doing all the tasks for daily living independently. Lisa marks her health as 'good'. However, Lisa lists several health problems, including carcinogenic neck lump and ongoing kidney problem with 4% kidney function restored (health status/ life experience): *I just had a nasty divorce (.) was trying to cope up (.) with the situation (.) you know it was hard, my daughter is very supportive (.) but dealing with the bereavement of my parents is harder […] I started having mental health problem (.) anxiety, depression […] was getting better with counselling (.) but had to stop.* Lisa feels very safe in her own home since she is there for 23 years (safety in the home). Therefore, she reveals that she is 'partially satisfied' in her life, mainly due to bereavement and financial problems (life satisfaction): *it's very difficult being on your own (.) hard to deal with the loneliness […] you know, at least someone to speak (.) when I am in need (.) in the middle of the night.* <u>Existing care and support</u>: Lisa started having counselling service for the last 3 months, and she felt that her condition was improving. However, she had to pay for the service independently, and it was very stressful for her. As Lisa mentions, *I stopped now because I couldn't afford it.* Lisa is 'fully satisfied' with her appearance and the food and drink she has in everyday life. But her needs are not fully met (care and support): *I have a good circle of friends (.) good family (.) still awful sometimes staying on my own (.) lack of emotional support […] especially at the weekends (.) Sundays (.) feel like to have someone.* <u>Concluding the interview with whether the interviewee wants to add anything more for his/her own care and support</u>: Lisa expresses her worry about her financial situation. Lisa experiences that there is a sort of gap between becoming single also reduces your income, because she does not have a partner who is financially supporting each other. Finally, she wants more accessible support or care (unmet need). <u>The result on unmet social care/support needs</u>: From the above narrative, it is noted that Lisa conveys her void of **loneliness** and **lack of emotional support** throughout her interview besides having **financial hardship**. She desires to have **more accessible support service**. Moreover, the privately paid counselling therapy was helping her but had to stop due to her financial hardship. In addition, she has a **lack of self-confidence obtaining existing social care services.** |
| 32 | **Alison (53 years, female, severely obese, positive cohabitation)** Alison is a 53-year-old lady, and lives with her partner. She is from a white British ethnic background and has severe obesity (BMI = 35.9 kg/m2). | <u>Life in terms of disability, health status and life satisfaction</u>: Alison faces several difficulties with the activities of daily living. Moreover, she is dependent on a walking stick; and sometimes she uses wheelchair and elbow crutches. In addition, she has a couple of health issues, mostly related to problems with her back and joints due to arthritis and osteoporosis. A few of her health conditions are long standing (health status). Alison marks 7 out of 10 (on a Likert scale of 0 to 10, where 10 is the best) regarding safety in her own home (safety in the home): *Mainly due to the stairs in the house (.) sometimes I feel dizzy (.) weak (.) sometimes I feel like (.) I might fall down one day.* Alison is 'partially satisfied' with her life overall (life satisfaction): |

*(Continued)*

**Table 2.** (Continued)

| Case No. | Name and demographics | Narrative analysis |
|---|---|---|
| | | *it varies and depends on the day (.) I have good days and bad days (.) but mostly bad (.) because lack of mobility makes me not so satisfied (.) I get depressed and frustrated (.) for being in pain (.) for taking so many pills (.) due to COVID, can't even go for swimming or other exercises.*<br><br>Existing care and support: Alison has a couple of housing adaptations like raised toilet and bath seat. However, she wants the bath to be taken out, as the shower will be more accessible. Alison had a few sessions of physio exercises that she did not have to pay. But she regrets that due to the pandemic, her physio sessions are cancelled. Alison's partner is her primary carer. Currently, Alison is receiving PEEP (Personal emergency evacuation plan) and reduced council tax. And her partner has just applied for a carer's allowance. The care and support she receives from her partner make her feel guilty (care and support/ unmet need):<br><br>*he is good (.) it's been a roller-coaster (.), but it's pretty good now […] sometimes struggle to make him realise the need is.*<br>She marks herself 'partially satisfied' in terms of the care and support she receives from her partner. However, Alison informs that she is happy about her partner's time and states that *sometimes I like to spend time on my own*. But she regards that her needs as 'hardly ever met'. According to her, *I don't feel the way I was before (.) can't do socialising.*<br>The way Alison's partner cares and looks after her makes her feel 'very good' about herself. However, the way others treat her makes her feel 'fair' about herself. As she conveys her feelings,<br>*He* (partner) *does try to understand […] he is getting older (.) he has medical needs too […] sometimes I feel like a burden […] a few months back, he fell ill (.) I was calm; it was difficult for me […] children came to rescue and looked after me […] family and friends don't understand me fully (.) my condition.*<br>She is 'fully satisfied' with her appearance and cleanliness. However, due to her food intolerance, it is difficult for her to get the right food.<br><br>Concluding the interview with whether the interviewee wants to add anything more for his/her own care and support:<br>Alison conveys her desire to do more exercise and considers that few more housing adaptations are needed to make her life easier to live (unmet need):<br>*The realisation from people or everyone (.) that I have good and bad days […] I want people's empathetic understanding towards me (.) not to decide by seeing my appearance and condition […] I don't want to be disabled (.) treated as disabled (.) trying to have a normal life.*<br>The result on unmet social care/support needs: From the above narrative, it is noted that Alison feels guilty for receiving care from her partner and perceives herself as a burden (**lack of emotional support**). She faces challenges in socialising (lack of socialisation) with others. In addition, she gets a sense of **social discrimination** while tries to mix with others. Moreover, Alison feels vulnerable due to a **lack of housing adaptations** and **lack of safety in her own house** due to the precarious nature of her existing unpaid care. In addition, Alison puts a high value on her independence to protect her self-esteem and misses doing her hobbies (swimming, exercise, physiotherapy) due to the pandemic. |
| 7 | **Ali (57 years, male, moderately obese, on his own)**<br>Ali is a 57-year-old widow and lives on his own. He is from a British Pakistani ethnic background and has moderate obesity (BMI = 32.4 kg/m2). | Life in terms of disability, health status and life satisfaction: Ali faces a problem using the toilet and using a bath as part of the difficulties in daily living. Although he does not use any type of aids to make his moving or daily living easier. Ali considers his general health is 'fair', mainly due to back pain and joint pain. However, he does not regard having any chronic illness (health status.<br>Ali assesses his safety score 5 out of 10 (on a Likert scale of 0 to 10, where 10 is the best), living in his own house (safety in the home).<br>Moreover, he interprets his overall life satisfaction as 'partially satisfied' (life satisfaction).<br>As Ali appraises his health and wellbeing:<br>*When I go to the bath, I have to be very careful (.) I couldn't stand for long […] I mean to say, there is not enough support to hold onto something […] I have slipped a couple of times (.) no walking shower (.) I do everything on my own […] there is no support (.) one lady came to visit me and said, she is going to write to my county or borough […] she wrote, but nothing happened […] I don't have any other health problems, but the back pain is the worst of all […] when I don't feel good, I try not to go out […] I were living in this flat* (Local Authority housing) *for the last 12 years […] I feel, people, come in my absence […] building entrance has a problem with lock […] lack of security […] because of my back pain, I lost my job (.) then all the problems started.*<br>Existing care and support: According to Ali, currently, he is not receiving any care or support, and his needs are 'hardly ever met'. Although he is 'fully satisfied' with his cleanliness and appearance. However, the foods and drink he gets in everyday life are not enough for him, making him 'unsatisfied' (care and support/ unmet need):<br>*I applied for the job seeker's allowances a few months back (.) but they refused (.) by saying that 'you didn't pay for the taxes, so you can't apply for this' […] if the employer doesn't give me enough shifts that I could pay the tax […] they only give me two days jobs […] it's not my fault [...] I pay for my national insurance number […] the situation is frustrating […] as I don't have any choice, managing as much as I can […] basically compromising.*<br>Concluding the interview with whether the interviewee wants to add anything more for his/her own care and support:<br>Ali expresses his desire to have a separate apartment with a walk-in shower (unmet need):<br>*if possible, to have proper support with few housing adaptations [...] as I live on my own […] help with daily living, such as the washing machine is in the distance […] even picking up something from the floor, makes me worried.*<br>The result on unmet social care/support needs: From the above narrative, it is noted that he is frustrated about his circumstances (**lack of emotional support**) besides having **financial hardships**. Moreover, Ali feels vulnerable due to **a lack of housing adaptations** and **safety in his own house.** His expressed needs are sub-optimally satisfied, as he does not qualify to access social care support (service-based). |

*(Continued)*

**Table 2.** (Continued)

| Case No. | Name and demographics | Narrative analysis |
|---|---|---|
| 19 | **Andrew (66 years, male, overweight, positive cohabitation and coresidents)** Andrew is a 66-year-old man and lives with his wife and daughter. He is from a white British ethnic background and is overweight (BMI = 28 kg/m2). | <u>Life in terms of disability, health status and life satisfaction</u>: Andrew encounters several challenges of daily living, such as using a map, recognising physical danger, preparing a hot meal, shopping for groceries, taking medication, managing money and sometimes with communication and making phone calls. Andrew gets confused and cannot remember everything. He does not like to use a walking stick, although he falls over a lot. And all these started after Andrew had a stroke in 2012, but Andrew considers his general health is good (Health status).<br>However, he does not feel very safe in his own home. And overall, Andrew perceives his life satisfaction as *very poor* (unsatisfied) (safety in the home). As Andrew examines his health and wellbeing:<br>*I can't remember what I am doing (.) sometimes I try to do something, I get confused (.) suppose I am making a cup of coffee (.) I can't remember how many spoonsful of coffee I have added (.) I just keep on adding […], then when I drink it, it tastes awful, and I get confused (.) I am sorry that I am explaining this […] I know I fall over a lot (.), but I keep saying to myself, I never ever going to use a wheelchair or anything like that (.) I probably sound stupid […] the days I feel unwell, I like to go out (.) I know it can be dangerous, but I have to keep trying (.) if I don't feel right, I just sit down a bit […] The days I am good, try to stay at home […] I get really upset.*<br><u>Existing care and support</u>: Andrew's wife and daughter look after him. And Andrew feels fully satisfied having the care and support from them. Andrew is happy about the time his wife and daughter spend with him. However, Andrew interprets that his needs are usually met. He is not so satisfied with his appearance and cleanliness and states, *I accept myself the way I am now.* However, Andrew is 'fully satisfied' with the foods and drink he gets in everyday life, and the care and support makes him feel excellent about himself (care and support/ unmet need):<br>*They* (wife and daughter) *are wonderful (.) but I can't remember if I get any other supports! […] everyone loves me now, obviously, that's the wonderful part of it (.) but most of the time (.) in my head (.) I get very upset to myself […] I wait for my wife or daughter to come back home from work for food […] I do try on my own, all I tend to take is (.) a Pizza (.), and I don't like it, but that takes 12 mins to microwave (.). If I don't have it, then I don't know what I am dealing with […] I look at egg (.) sometimes I feel like to have it (.), but I can't remember what to do with it (.) then I get confused […] So even I don't like Pizza, I still have it […] at least I got something to eat.*<br><u>Concluding the interview with whether the interviewee wants to add anything more for his/her own care and support</u>: Andrew expresses that he cannot follow a conversation if in a group but feels like contributing during a one-to-one conversation. Andrews accepts that it is difficult for someone to look after him, but thankful for his wonderful family, who is always ready to serve him. Andrew loves mixing with people and loves to share his experiences. Although he cannot remember everything, he loves helping other people in need (unmet need).<br><u>The result on unmet social care/support needs</u>: From the above narrative, it is noted that Andrew has several existing needs in terms of ADLs and IADLs, but lack of support with everyday activities necessitates. Although he is having some supports from his family (informal carers), he is somehow neglected (**have inadequate existing care**) due to their (carer's) busy schedules (**lack of carer's time**). Moreover, Andrew tries to compromise with his present situation and expresses his frustration (**lack of emotional support**). In addition, he is vulnerable in his own home in terms of his foods and drinks, staying on his own for a long time (**lack of safety in his own house**). In addition, Andrew puts a high value on his independence to protect his self-esteem and misses being independent. And he feels his self-worth by getting the chance to help others. |
| 21 | **Kevin (73 years, male, moderately obese, on his own)** Kevin is a 73-year-old widow and lives on his own. He is from a white British ethnic background and has moderate obesity (BMI = 32.8 kg/m2). | <u>Life in terms of disability, health status and life satisfaction</u>: Despite having a heavy-weight, Kevin is entirely self-sufficient and capable of doing all the tasks for daily living independently and considers his general health 'good'. Although he had his double knee replacement 6 years ago and existing arthritis with painful fingers (health status).<br>He feels reasonably safe in his own house and scores it 9 out of 10 (where 10 is the safest), and fully satisfied with his life overall (safety in the home).<br>*Since then, although I don't have any problem* (following knee replacement) *[…] I find that I can't run (.) and I like to say, I can't kneel down (.) and also (.) once I used to do my own decorating, I would kneel down and paint the skirting and all that (.) but now I can't do anymore (.) that's the only drawback […] I feel quite safe, but always that possibility that something happens (.) in my back of mind (.) as no one is there.*<br><u>Existing care and support</u>: Kevin does not receive any form of social care support. He is 'fully satisfied' with his appearance and the foods and drinks he has in everyday life; besides, the way everyone treats him makes him feel 'very good' about himself (not excellent). However, all his needs are not met fully, as he mentions 'usually met' (care and support/ unmet need):<br>*Obviously, with this circumstance* (COVID-19 pandemic) *now, I receive plenty of offers from my neighbours […] if I need any shopping or anything! (.) apart from that, I do all myself […] my wife passed away 6 years ago […] couple of years later, I applied for a part-time job (.) taking special need children to and from school […] so that's get me out of the house […] obviously, interacting with people keeps me busy […] obviously, during the evening, I am on my own […] that's why I do this driving job […] that keeps me busy and (.) my mind from going crazy […] my sons are away (.) they are busy in their own life […] I always keep in touch with my family members […] obviously, before my wife passed away, I mean I walked in from work, and my meals were on the table […] I have to do all that myself now! (.) and I decided, I don't want to live on fast food […] so, I try to make a balanced diet, something different every day.*<br><u>Concluding the interview with whether the interviewee wants to add anything more for his/her own care and support</u>: Kevin ends the conversation by saying that he is reasonably *content* and happy to live independently (unmet needs):<br>*Don't need anything more […] like I said, there is always someone there, at the other end of the telephone […] I am happy with my life.* |

*(Continued)*

**Table 2.** (Continued)

| Case No. | Name and demographics | Narrative analysis |
|---|---|---|
| | | <u>The result on unmet social care/support needs</u>: Kevin misses out on doing his old hobby (decorating) and his run, as his health condition does not permit that anymore (**lack of emotional support**). Moreover, living alone makes him paranoid about his situation that someday something might get wrong (**lack of emotional support**). In addition, Kevin is reluctant to request help and **hesitant to accept having unmet needs.** But Kevin feels coded by getting the chance to do part-time work and puts a high value on his independence. However, Kevin admits that he does this job mainly to protect his mental health by being able to engage his mind and being able to socialise with others (*obviously, during the evening, I am on my own […] that's why I do this driving job […] that keeps me busy and (.) my mind from going crazy*) (to avoid being **lonely** and **socially isolated**). |
| 14 | **Daniel (51 years, male, severely obese, positive cohabitation)** Daniel is a 51-year-old male and lives with his wife. He is from a white Spanish ethnic background and has severe obesity (BMI = 39.6 kg/m2). | <u>Life in terms of disability, health status and life satisfaction</u>: Despite having a heavy-weight, Daniel is entirely self-sufficient and capable of doing all the tasks for daily living independently and considers his general health very good (health status). Moreover, Daniel feels very safe in his own house. Although he has gout with very painful left foot and knees (safety in the home). *Somehow my energy is drained (.) I have sleep apnoea (.) I think the actual problem is there […] I am doing lots of exercises, but not lost enough weight […] sometimes I stop walking or exercise, because of gout […] I feel frustrated […] not losing weight regardless of doing exercise.* <u>Existing care and support</u>: Daniel feels 'fully satisfied' with the support he receives from his wife. Besides, Daniel is pleased with his wife's time with him and 'fully satisfied' with his appearance and the food and drinks he has in everyday life. Moreover, Daniel judges that all his needs are 'fully met' and feels 'very good' about himself how his wife and others treat him. As he examines his care needs (care and support/ unmet need): *My wife is always there to help (.) but I try not to ask too much, try to be independent […] sometimes our opinion differs […] I like to do things in a particular way (.) want to listen to a lot of music (.), but she doesn't like […] it's too noisy for her and (.) I have to always use a headphone […], not everything is perfect, there is always a room for improvement […] looking the way some people live (.) I live comfortably, but if there is money (.) I could live more comfortably.* <u>Concluding the interview with whether the interviewee wants to add anything more for his/her own care and support</u>: Daniel ends the interview by thanking me for giving him the opportunity and time to express his views and says *I hope the study helps to get things right.* <u>The result on unmet social care/support needs</u>: Daniel wishes to be fitter and lose weight but gets disappointed and frustrated (**lack of emotional support**) for not achieving weight loss as he desires. Besides that, although his wife is supportive, sometimes the difference of opinions makes him annoyed (**lack of emotional support**). Moreover, Daniel is reluctant to request help and **hesitant to accept having unmet needs.** Financially Daniel is stable, but he feels upset while comparing himself with others (**lack of emotional support**). |

lack of housing adaptations and safety, boredom, inability to continue hobbies, and limited access to support. In contrast, normal-weight and overweight participants (cases 3, 19) identified gaps in service provision, such as insufficient carer time and knowledge.

Narratives also highlight that past health conditions (cases 21, 14), mobility challenges (cases 3, 7, 19, 32, 30, 27, 2), or significant life events like bereavement and divorce (case 10) contribute to unmet needs. Older adults living with a spouse, partner, or children generally receive more care than those living alone. It also shows how the sense of frustration, social discrimination, lack of life purpose, depression, being a burden, guilt, embarrassment, anxiety, boredom, and being unable to do things they used to do diminish participants' self-worth and self-confidence. In addition, the narratives reveal that despite having a sound support system in terms of money, housing adaptations, mobility aids and unpaid care support (case number-27) or not having any difficulty in daily living (case number- 21, 14) the older adult participants can still experience low self-esteem.

It is also noted that loneliness and social isolation are more critical issues for many older adults (case number- 2, 10, 32, 21) than actual physical needs regardless of their disability and health status.

Some older adults feared social discrimination (cases 27, 32), judgment (cases 27, 32), and anxiety and panic (case 30) along with other contributing factors, leading to isolation and reduced wellbeing. For many older adults, relying on a spouse for care and support (culturally unacceptable in some cases, e.g., case 3) is a key factor contributing to poor well-being (cases

27, 30, 32). Narratives suggest that those with lower life satisfaction are more likely to experience poor wellbeing and unmet needs, regardless of the support they receive (cases 2, 30, 10, 32, 7). Daniel (case 14) feels frustrated over not losing weight and hesitates to acknowledge unmet needs, though his lack of emotional support emerges as a concern. His sleep apnoea, linked to a high BMI [2,35], also contributes to fatigue and low energy.

Conversely, some older adults report life satisfaction despite unstable care support. Those engaged in hobbies (case 32, though Alison struggles due to the pandemic), helping others (cases 19, 21), working part/full-time (cases 21, 2; though Sunita feels isolated after losing work due to the pandemic), maintaining social connections (even via phone, case 21), or having a positive cohabitation relationship (case 14) tend to experience better wellbeing.

It is also noted that financial hardship is another unmet need that older adults are having. For example, Ali (case number 3) was in full-time employment, but he lost his job due to his painful back, and joints resulting from his obesity. However, older adults who put a high value on their independence to protect their self-esteem and hold a positive outlook are deemed to have higher wellbeing regardless of their increasing weight and also seem to have fewer unmet care needs than others (case number- 21, 27). The above narratives, also reveal that in many cases, minimal support like few adaptations or aids or just providing a counselling service (case number- 10) and/or providing a social network may help older adults to meet their needs.

## Discussion

The narrative analyses of the qualitative semi-structured interview indicate that participants face challenges in their everyday lives regardless of their weight categories. Moreover, the stories reveal that older adults with disabilities concerning difficulties in either ADLs, IADLs and/or mobilities struggle to meet a range of compound needs than other older adult participants. The findings are supported by Age UK [13] that found living with a single 'need' can be as stressful as living with compound needs and that if they remain unmet, then individuals are often struggling with a combination of unmet needs. Several past studies have found that various types of arthritis, cancer, and kidney disease can be directly or indirectly associated with individuals' high BMI level [36–38]. This is also reflected in the narrative stories of the study participants that the main reason for their disability is either due to their painful back and joints or the pain due to cancer, kidney dysfunction and previous surgery.

Moreover, the narratives of the present study explore that loneliness and social isolation are critical issues for many older adults regardless of their disability and health status. Yet loneliness is a key predictor of poor self-esteem and lack of self-confidence [39]. A longitudinal study on English older adults by Shankar et al [40] explored that loneliness and social isolation are associated with an individual's poor wellbeing. These findings are also in line with a NatCen study by Dunatchik et al [41] from a primary interview dataset of older adults that "older people raised unmet need for social contact and mobility as being as important, if not more important as meeting basic needs of daily living". Furthermore, several studies support the finding that high BMI increases the risks/likelihood of loneliness, social discrimination, being socially judged and social anxiety [2,42,43].

Furthermore, it is noted from the interview narratives that some older adults are more satisfied in their lives overall, despite their unstable care and support system. Age UK [13] states that "wellbeing for older people is multi-faceted and includes health, care and support, money, housing and social contact. It follows that people living with disadvantage in these areas, and even with a single need, are more likely to have lower wellbeing". A longitudinal study on the secondary ELSA dataset [20] found that older adults' level of wellbeing is not a significant predictor of their future unmet care needs. Another report [41] on a primary interview dataset of older adults showed that there are some areas of unmet care need that have a stronger

association with wellbeing than others, these are "lack of mobility and isolation and lack of access to hobbies and interests and the associated loss of independence". However, both studies considered the study participants included older adults that may be underweight and malnourished and so may not wholly apply to overweight and older adults with obesity. At the same time, the study by Dunatchik et al [41] found that an easily accessible helpline or community centres can be an intervention that reduces the risk of negative wellbeing. This is in line with this present study's findings that many older adults are frustrated and upset due to a lack of easily accessible support or helpline (case number- 2, 10). It is also evident that implementing effective interventions, such as home modifications and assistive technologies, can help older adults maintain independence and reduce future demand for social care services [20].

Financial hardship is found to be another crucial issue for many older adults in the present study since there is an association between obesity and unemployment [2], and older adults experiencing major transitional phases of life, like retirement [1]. As such, a cross-sectional study by Conklin et al. [44] established obesity demands financial hardship regardless of social class, education, and housing tenure. However, older adults who put a high value on their independence to protect their self-esteem and hold a positive outlook are deemed to have higher wellbeing. Although, this finding needs to be treated with caution as according to Dunatchik et al [41], "managing to cope, but with impacts on exhaustion and pain, or by limiting expectations is an indication of unmet need".

It is, however, important to state that qualitative interviews reveal people's perspectives to see their lives rather than quantify the number of people holding those outlooks [41].

## Conclusion

The study conceptualises unmet care needs as gaps in service provision, including activity-based needs (e.g., ADLs, IADLs) and broader psychosocial factors such as loneliness, safety concerns, and social support. Unmet care needs were classified as either unexpressed (participants unaware of their needs or reluctant to seek support) or expressed but sub-optimally met due to eligibility criteria or service limitations. The findings contribute to the broader discourse on care provision, emphasising the necessity of addressing both practical and psychosocial dimensions of support for older adults. Many participants living alone or lacking strong social networks experienced heightened feelings of vulnerability and isolation. The key emerging themes included: the impact of chronic pain and disability on daily functioning, social isolation as a predictor of low self-esteem and emotional distress, financial hardship as a barrier to accessing necessary care, the role of informal caregivers and its cultural implications and the significance of minimal social support interventions in improving wellbeing. Policy-makers and healthcare providers should consider these insights to develop integrated obesity management strategies that encompass both medical and social care dimensions in a more holistic way. Future research should further explore targeted interventions that empower older adults and enhance their quality of life.

## Supporting information

**S1 File. Consent form.**
(DOCX)

**S2 File. Data management and storage statement.**
(DOCX)

**S3 File. Ethical approval letters.**
(DOCX)

**S4 File. General practitioner information sheet.**
(DOCX)

**S5 File. Participants information sheet.**
(DOCX)

**S6 File. Semi-structured questionnaires for qualitative interviews.**
(DOCX)

## Acknowledgments

This research is an outcome of the first author's PhD study at the University of West London, UK.

## Author contributions

**Conceptualization:** Gargi Ghosh.

**Data curation:** Gargi Ghosh.

**Formal analysis:** Gargi Ghosh.

**Investigation:** Gargi Ghosh.

**Methodology:** Gargi Ghosh.

**Project administration:** Gargi Ghosh.

**Supervision:** Hafiz T. A. Khan, Salim Vohra.

**Validation:** Hafiz T. A. Khan.

**Visualization:** Gargi Ghosh.

**Writing – original draft:** Gargi Ghosh.

**Writing – review & editing:** Gargi Ghosh, Hafiz T. A. Khan, Salim Vohra.

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
