## [Decision Letter · Decision Letter 0]

21 Oct 2024

PONE-D-24-21647Factors Influencing the Hidden Burden of Care for Older Adults, with Overweight and Obesity, in England: A Qualitative StudyPLOS ONE

Dear Dr. Ghosh,

Thank you for submitting your manuscript to PLOS ONE. After careful consideration, we feel that it has merit but does not fully meet PLOS ONE’s publication criteria as it currently stands. Therefore, we invite you to submit a revised version of the manuscript that addresses the points raised during the review process.

We look forward to receiving your revised manuscript.

Kind regards,

Anteneh Mengist Dessie, MPH

Academic Editor

PLOS ONE

**Journal Requirements:**

2. We note that this data set consists of interview transcripts. Can you please confirm that all participants gave consent for interview transcript to be published?

If they DID provide consent for these transcripts to be published, please also confirm that the transcripts do not contain any potentially identifying information (or let us know if the participants consented to having their personal details published and made publicly available). We consider the following details to be identifying information:

- Names, nicknames, and initials

- Age more specific than round numbers

- GPS coordinates, physical addresses, IP addresses, email addresses

- Information in small sample sizes (e.g. 40 students from X class in X year at X university)

- Specific dates (e.g. visit dates, interview dates)

- ID numbers

Or, if the participants DID NOT provide consent for these transcripts to be published:

- Provide a de-identified version of the data or excerpts of interview responses

- Provide information regarding how these transcripts can be accessed by researchers who meet the criteria for access to confidential data, including:

a) the grounds for restriction

b) the name of the ethics committee, Institutional Review Board, or third-party organization that is imposing sharing restrictions on the data

c) a non-author, institutional point of contact that is able to field data access queries, in the interest of maintaining long-term data accessibility.

d) Any relevant data set names, URLs, DOIs, etc. that an independent researcher would need in order to request your minimal data set.

For further information on sharing data that contains sensitive participant information, please see: https://journals.plos.org/plosone/s/data-availability#loc-human-research-participant-data-and-other-sensitive-data

If there are ethical, legal, or third-party restrictions upon your dataset, you must provide all of the following details (https://journals.plos.org/plosone/s/data-availability#loc-acceptable-data-access-restrictions):

a) A complete description of the dataset

b) The nature of the restrictions upon the data (ethical, legal, or owned by a third party) and the reasoning behind them

c) The full name of the body imposing the restrictions upon your dataset (ethics committee, institution, data access committee, etc)

d) If the data are owned by a third party, confirmation of whether the authors received any special privileges in accessing the data that other researchers would not have

e) Direct, non-author contact information (preferably email) for the body imposing the restrictions upon the data, to which data access requests can be sent

Reviewers' comments:

Reviewer's Responses to Questions

**Comments to the Author**

1. Is the manuscript technically sound, and do the data support the conclusions?

Reviewer #1: Yes

Reviewer #2: Yes

2. Has the statistical analysis been performed appropriately and rigorously? 

Reviewer #1: N/A

Reviewer #2: Yes

3. Have the authors made all data underlying the findings in their manuscript fully available?

Reviewer #1: Yes

Reviewer #2: Yes

4. Is the manuscript presented in an intelligible fashion and written in standard English?

Reviewer #1: Yes

Reviewer #2: Yes

5. Review Comments to the Author

**Reviewer #1:**  It is a very good attempt. I really like the way everything has been mentioned in detail. Methodology is extremely vigorous. Results have been explained very clearly. Results have been explained very clearly.

**Reviewer #2:**  Factors Influencing the Hidden Burden of Care for Older Adults, with Overweight and Obesity, in England: A Qualitative Study is an interesting topic and noble cause to find out the responsible factors for the caring older adults with obesity in Britain society. I will suggest few changes in the manuscript before publication. These changes are following: Write full spellings of abbreviations used in abstract and font issues, and repeated full stops (Mentioned with yellow highlight in the pdf file). Need to short and avoid repeating the methodology section.

Page # 02: full spellings of NHS.

Page # 02: were all older participants (male or female or both) in this study, have British background or immigrant to British from different nationalities? Mentioned in Abstract. As later on mentioned in Table 1.

Page # 14: Figure 1 is not shown in PDF file.

Table 2 can put all data in supplementary files.

6. PLOS authors have the option to publish the peer review history of their article (what does this mean? ). If published, this will include your full peer review and any attached files.

**Do you want your identity to be public for this peer review?** For information about this choice, including consent withdrawal, please see our Privacy Policy .

Reviewer #1: No

Reviewer #2: No

---

## [Author Response · Author response to Decision Letter 1]

30 Oct 2024

1. have changed the heading and sub-heading styles, title page, in text citation & referencing style, fig & table requirements and supporting documents, as requested.

2. The ethical statement section under methods has been amended as recommended. Included the statement- “All participants provided written and signed consent prior to participation by completing the informed consent forms. The participants also consented to use their interview transcripts, with all identifiable details removed, for teaching, future research, and publication purposes. It is confirmed that the transcripts do not contain any identifying information. All participants in this study were assigned pseudonyms, with their interview data organised and anonymised using sequential case numbers for analysis”

3. I have shortened the methods section by shortening the ‘Underpinning theory’ section.

4. I have corrected the repetitive statements.

5. I have modified all the abbreviations by the full spelling once (e.g, National Health Service for NHS)

6. I have modified the abstract as requested with a sentence- “All participants in this study are either of British origin or immigrants to the UK from various nationalities.”

6. The author plans to retain Table 2 within the main body of the text, as both the results and discussions are directly tied to it. The author believes that without this table, it would be challenging for readers to fully grasp the study's findings and discussions.

7. I have changed the abstract with the correct one to keep it aligned with the Manuscript.

---

## [Decision Letter · Decision Letter 1]

8 Jan 2025

PONE-D-24-21647R1Factors Influencing the Hidden Burden of Care for Older Adults, with Overweight and Obesity, in England: A Qualitative StudyPLOS ONE

Dear Dr. Ghosh,

Thank you for submitting your manuscript to PLOS ONE. After careful consideration, we feel that it has merit but does not fully meet PLOS ONE’s publication criteria as it currently stands. Therefore, we invite you to submit a revised version of the manuscript that addresses the points raised during the review process.

1. Objective of the Study: It is written as quantitative using verb “to examine the association between …” Qualitative studies are not intended for examining the association rather focus on exploration and understanding the phenomena. Revise the study objective to better align with qualitative research language.

2. Ethical Consideration: Simplify the ethical consideration section to retain essential details while reducing verbosity.

3. Abstract: Provide a concise background that highlights the research gap in 1-2 line.

4. Introduction: It is very lengthy and difficult to follow. Streamline the introduction by focusing on the key background, research gap, and objectives in a concise manner.

5. Methods: All sections are overly detailed, resembling a thesis. Please revise to present the content concisely, particularly in the ethical aspect and underpinning theory sections. Condense sections while maintaining clarity.

Underpinning Theory: Limit to 1–2 paragraphs with only relevant points for this study, omitting extensive theoretical details. Discussion of underpinning theory should be condensed into a few sentences that directly tie to the study.

Eligibility: Combine bullet points into a paragraph

Try to avoid using terms "measurement of variables" like quantitative study.

6. Conclusion: Focus on summarizing the study's findings and implications in a concise manner, avoiding unrelated comparisons (line 4) like discussion. Avoid using terms like "outcomes" in the context of qualitative research.

7. Overall Manuscript: Throughout the manuscript, prioritize brevity by including only key information directly relevant to the study. Remove excessive elaboration (e.g., thesis-level details). Ensure a clear flow by presenting information systematically, avoiding redundancy.

We look forward to receiving your revised manuscript.

Kind regards,

Dr Buna Bhandari

Academic Editor

PLOS ONE

Journal Requirements:

Additional Editor Comments:

1. Objective of the Study: It is written as quantitative using verb “to examine the association between …” Qualitative studies are not intended for examining the association rather focus on exploration and understanding the phenomena. Revise the study objective to better align with qualitative research language.

2. Ethical Consideration: Simplify the ethical consideration section to retain essential details while reducing verbosity.

3. Abstract: Provide a concise background that highlights the research gap in 1-2 line.

4. Introduction: It is very lengthy and difficult to follow. Streamline the introduction by focusing on the key background, research gap, and objectives in a concise manner.

5. Methods: All sections are overly detailed, resembling a thesis. Please revise to present the content concisely, particularly in the ethical aspect and underpinning theory sections. Condense sections while maintaining clarity.

Underpinning Theory: Limit to 1–2 paragraphs with only relevant points for this study, omitting extensive theoretical details. Discussion of underpinning theory should be condensed into a few sentences that directly tie to the study.

Eligibility: Combine bullet points into a paragraph

Try to avoid using terms "measurement of variables" like quantitative study.

6. Conclusion: Focus on summarizing the study's findings and implications in a concise manner, avoiding unrelated comparisons (line 4) like discussion. Avoid using terms like "outcomes" in the context of qualitative research.

7. Overall Manuscript: Throughout the manuscript, prioritize brevity by including only key information directly relevant to the study. Remove excessive elaboration (e.g., thesis-level details). Ensure a clear flow by presenting information systematically, avoiding redundancy.

Reviewers' comments:

Reviewer's Responses to Questions

**Comments to the Author**

1. If the authors have adequately addressed your comments raised in a previous round of review and you feel that this manuscript is now acceptable for publication, you may indicate that here to bypass the “Comments to the Author” section, enter your conflict of interest statement in the “Confidential to Editor” section, and submit your "Accept" recommendation.

Reviewer #1: All comments have been addressed

Reviewer #2: All comments have been addressed

2. Is the manuscript technically sound, and do the data support the conclusions?

Reviewer #1: Yes

Reviewer #2: Yes

3. Has the statistical analysis been performed appropriately and rigorously? 

Reviewer #1: Yes

Reviewer #2: Yes

4. Have the authors made all data underlying the findings in their manuscript fully available?

Reviewer #1: Yes

Reviewer #2: Yes

5. Is the manuscript presented in an intelligible fashion and written in standard English?

Reviewer #1: Yes

Reviewer #2: Yes

6. Review Comments to the Author

Reviewer #1: All the revisions have been done now. The paper is now good to be published. There is no problem whatsoever regarding the technical aspects of the paper. I would like to see the paper published as such.

Reviewer #2: I reviewed the revisions of article "Factors Influencing the Hidden Burden of Care for Older Adults, with Overweight and Obesity, in England: A Qualitative Study" and found authors have been addressed all comments.

7. PLOS authors have the option to publish the peer review history of their article (what does this mean? ). If published, this will include your full peer review and any attached files.

**Do you want your identity to be public for this peer review?** For information about this choice, including consent withdrawal, please see our Privacy Policy .

Reviewer #1: No

Reviewer #2: No

---

## [Author Response · Author response to Decision Letter 2]

11 Feb 2025

Dear team,

Thank you for reviewing this article and providing your valuable feedback. Following the guidance from the academic editors and two reviewers, I have revised and resubmitted the article. Updates have been made to the title, abstract, introduction, methods, conclusion, and the entire document as requested. Additionally, I have incorporated the suggested corrections. The revised document has been uploaded as a response to the reviewers and is also attached below.

Thank you.

Kind regards,

Gargi

Reviewer and editorial comments Amendment details The page number on the Manuscript (if any)

1. Objective of the Study: It is written as quantitative using verb “to examine the association between …” Qualitative studies are not intended for examining the association rather focus on exploration and understanding the phenomena. Revise the study objective to better align with qualitative research language. Amended the study objective as recommended

pp-2 & 5

2. Ethical Consideration: Simplify the ethical consideration section to retain essential details while reducing verbosity. Amended as recommended

PP-7

3. Abstract: Provide a concise background that highlights the research gap in 1-2 line. A concise background is provided highlighting the research gap PP-2

4. Introduction: It is very lengthy and difficult to follow. Streamline the introduction by focusing on the key background, research gap, and objectives in a concise manner.

I have revised the introduction as recommended, reducing its length from four pages to two and a half pages. pp-4, 5 & 6

5. Methods: All sections are overly detailed, resembling a thesis. Please revise to present the content concisely, particularly in the ethical aspect and underpinning theory sections. Condense sections while maintaining clarity.

Underpinning Theory: Limit to 1–2 paragraphs with only relevant points for this study, omitting extensive theoretical details. Discussion of underpinning theory should be condensed into a few sentences that directly tie to the study.

Eligibility: Combine bullet points into a paragraph

Try to avoid using terms " like quantitative study. I have revised the methodology section, including the ethical aspects, theory, and eligibility, as recommended. The theory section is now two paragraphs long. For eligibility, I merged the bullet points into a single paragraph. Additionally, terms such as "measurement of variables" have been removed. pp- 6, 7 & 10

6. Conclusion: Focus on summarizing the study's findings and implications in a concise manner, avoiding unrelated comparisons (line 4) like discussion. Avoid using terms like "outcomes" in the context of qualitative research. I have revised the conclusion as recommended. PP-33 & 34

7. Overall Manuscript: Throughout the manuscript, prioritize brevity by including only key information directly relevant to the study. Remove excessive elaboration (e.g., thesis-level details). Ensure a clear flow by presenting information systematically, avoiding redundancy. I have thoroughly revised the manuscript as recommended, ensuring a logical flow by presenting information systematically and eliminating redundancy.

---

## [Editor Report · Decision Letter 2]

16 Feb 2025

A Qualitative study to examine hidden care burden for older adults with overweight and obesity in England

PONE-D-24-21647R2

Dear Dr. Ghosh,

We’re pleased to inform you that your manuscript has been judged scientifically suitable for publication and will be formally accepted for publication once it meets all outstanding technical requirements.

Kind regards,

Dr Buna Bhandari

Academic Editor

PLOS ONE
---

## [Editor Report · Acceptance letter]

PONE-D-24-21647R2

PLOS ONE

Dear Dr. Ghosh,

I'm pleased to inform you that your manuscript has been deemed suitable for publication in PLOS ONE. Congratulations! Your manuscript is now being handed over to our production team.

Kind regards,

on behalf of

Dr. Buna Bhandari

Academic Editor

PLOS ONE